# Hands off the Mink! Using Environmental Sampling for SARS-CoV-2 Surveillance in American Mink

**DOI:** 10.3390/ijerph20021248

**Published:** 2023-01-10

**Authors:** Ellen Boyd, Michelle Coombe, Natalie Prystajecky, Jessica M. Caleta, Inna Sekirov, John Tyson, Chelsea Himsworth

**Affiliations:** 1Ministry of Agriculture and Food, Government of British Columbia, Abbotsford, BC V3G 2M3, Canada; 2School of Population and Public Health, University of British Columbia, Vancouver, BC V6T 1Z3, Canada; 3BC Centre for Disease Control, Vancouver, BC V5Z 4R4, Canada

**Keywords:** SARS-CoV-2, American mink, environmental surveillance

## Abstract

Throughout the COVID-19 pandemic, numerous non-human species were shown to be susceptible to natural infection by SARS-CoV-2, including farmed American mink. Once infected, American mink can transfer the virus from mink to human and mink to mink, resulting in a high rate of viral mutation. Therefore, outbreak surveillance on American mink farms is imperative for both mink and human health. Historically, disease surveillance on mink farms has consisted of a combination of mortality and live animal sampling; however, these methodologies have significant limitations. This study compared PCR testing of both deceased and live animal samples to environmental samples on an active outbreak premise, to determine the utility of environmental sampling. Environmental sampling mirrored trends in both deceased and live animal sampling in terms of percent positivity and appeared more sensitive in some low-prevalence instances. PCR CT values of environmental samples were significantly different from live animal samples’ CT values and were consistently high (mean CT = 36.2), likely indicating a low amount of viral RNA in the samples. There is compelling evidence in favour of environmental sampling for the purpose of disease surveillance, specifically as an early warning tool for SARS-CoV-2; however, further work is needed to ultimately determine whether environmental samples are viable sources for molecular epidemiology investigations.

## 1. Introduction

The SARS-CoV-2 pandemic has shown the usefulness of environmental sampling in disease surveillance and outbreak response [1,2]. Environmental sampling can provide a snapshot of disease at the population level without having to test many individuals. It becomes especially useful once proven to mirror or predict the results from more conventional screening programs (i.e., individual PCR tests for COVID-19) in terms of incidence rates and allows for a more cost-effective and less labour-intensive surveillance [1,2,3]. This concept has been demonstrated through wastewater sampling for COVID-19, where wastewater signals preceded confirmed cases by up to 63 days, with 13 studies reporting positives prior to any detection within the community [1]. Wastewater signals were associated with viral loads within community cases in many instances, allowing for efficient surveillance [1,2]. The importance of herd health (i.e., controlling or eliminating disease and management inefficiencies that impact animal welfare) within agriculture means that environmental sampling has the potential to be a valuable tool for surveillance in livestock species.

Throughout the COVID-19 pandemic, numerous non-human species were shown to be susceptible to natural infection with SARS-CoV-2, including many domesticated species [4]. Farmed American mink (*Neovison vison*) are one of the most highly susceptible non-human species, with evidence of human to mink, mink to human and mink to mink transmission [5,6,7,8]. Once infected, American mink have a high rate of viral mutation, increasing the likelihood of producing variants that are more virulent, or vaccine-resistant [5,6,9,10]. This poses significant health risks to the farmed mink themselves and to the humans who work closely with them and to those in nearby communities. Cases have been documented of mink-specific viral mutations identified within communities in the proximity of mink farms, but with no direct contact with mink [5]. Due to the possible public health risks, ongoing surveillance of active SARS-CoV-2 outbreaks on mink farms is imperative. Surveillance may also be needed to track the progression of an outbreak on a mink farm, particularly where infected farms are managed through quarantine, rather than depopulation [11].

SARS-CoV-2 surveillance within mink farms is historically conducted through the testing of recent mortalities and/or sampling of live mink [12]. However, both methodologies have significant limitations. Mortality testing is problematic as American mink tend to have highly variable degrees of morbidity and mortality once infected, with some farms seeing mass die-offs and others experiencing almost no losses [13]. Given the fractious nature of most American mink and the dive reflex, which enables animals to seal their nares, live sampling (nasopharynx or oropharyngeal swabbing) is very challenging and labour-intensive [14,15]. Furthermore, clinical signs of SARS-CoV-2 infection in American mink are variable, with 30% of farmers in one study reporting no clinical signs despite active infections in their herds [13]. This means that targeting symptomatic animals for the purposes of disease detection may not be feasible, requiring more systematic sampling schemes involving large numbers of live animals to determine the presence or absence of disease with high confidence [16]. Previous studies have evaluated environmental samples such as air, food, water, food residues, and faecal material [7,10,16,17]. The detection of SARS-CoV-2 in these sampling methods has been variable.

The purpose of this study was to continue the evaluation of the utility of environmental sampling for SARS-CoV-2 surveillance in mink. Specifically, we aimed to address the following research questions:How do results from environmental samples compare to those obtained from live and dead mink in terms of percent positivity and PCR CT values?In terms of environmental sampling methods, how do manure trough swabs or cage swabs compare regarding detection of SARS-CoV-2?Can environmental samples pick up sufficient viral RNA for successful genomic sequencing to be used in molecular epidemiological studies (i.e., variant characterization)?

## 2. Materials and Methods

### 2.1. Study Location

This study took place during an outbreak of SARS-CoV-2 on a mink farm in British Columbia, Canada. The outbreak was detected on 2 May 2021, and continued until April 2022, at which point the farm was mandated to be depopulated by the BC Ministry of Agriculture and Food. The outbreak was detected during a mandatory passive surveillance of all dead mink on BC farms—a program that was initiated after the first outbreak of SARS-CoV-2 on a different farm on 21 April 2021.

### 2.2. Mortality Surveillance

All mink that died on-farm were frozen and submitted weekly to the Animal Health Centre (AHC), BC Ministry of Agriculture and Food. Mink carcasses were tested once per week. Carcasses were thawed and a cross-sectional cut was made at the level of the cartilaginous nasal septum. A sterile 1.5 mL swab was inserted into the exposed nasal canal to the level of the notched indicator on the swab so that the tip of the swab reached the back of the nasopharynx. The swab was swirled several times and then placed in a tube containing viral transport media (VTM).

### 2.3. Live Animal Surveillance

Approximately bi-weekly, a stratified random selection of 65 live mink was tested in accordance with BC Ministry of Agriculture and Food protocols to detect the presence of SARS-CoV-2 at a ≥5% prevalence. Each mink was selected at random with approximately evenly distributed locations through the sheds to represent the spatial extent of the farm and the approximate age–sex distribution of the farm. For instance, if there were 13 active mink sheds, then 1 mink was selected at random from five approximately evenly spaced regions within each shed. If 33% of the mink herd was adult female, then every third mink selected would be an adult female. Oropharyngeal samples were collected from each selected mink by the herd veterinarian. The mink was manually restrained, and a sterile polyester swab was inserted into the mink’s mouth through a gag (either a 3cc syringe tube or a short length of sterilised 1 inch diameter copper pipe for adult male mink). The swab was twirled several times as far back in the oral cavity as possible and then placed into a pre-labelled tube containing VTM.

### 2.4. Environmental Sampling

Environmental samples were taken at the same time as live animal samples. While the mink was removed from its cage for an oral sample, a paired cage sample was also taken. The cage was sampled by rubbing a sterile polyester swab along an approximately 10 cm × 10 cm area of the wire mesh surface of both the cage ceiling and floor. Mink feed is placed on top of the enclosures and their manure drops into a collection system through the mesh flooring. Therefore, these areas of the cage were expected to have a higher likelihood of exposure to bodily fluids (saliva, mucous, nasal excretions, faeces, or urine) that contain SARS-CoV-2.

Manure troughs are an automated manure collection system with one trough collecting all urine and faeces, under a row of multiple mink cages. These troughs provide a convenient sample encompassing excreta from many individuals. The manure troughs (n = 12) were sampled by inserting a sterile polyester swab into the outflow end of the trough, swirling the swab several times, and then placing the swab into a pre-labelled tube containing VTM.

Environmental sampling continued every two weeks for a total of 6 weeks after the mink were depopulated, at which time the farm was decommissioned.

### 2.5. Sample Testing

#### 2.5.1. SARS-CoV-2 PCR Testing at the Animal Health Center

The majority of mortality samples were tested at the Animal Health Center using qPCR. Viral RNA was isolated using the Applied Biosystems Incorporated (ABI) MagMax-96 Express magnetic particle processor (ThermoFisher Scientific, Waltham, MA, USA) with the MagMax™-96 Viral RNA Isolation Kit (ThermoFisher, catalog number: AM1836), as per kit instructions. The MagMax program (“AM1836_DW_v50”) was available on the ThermoFisher website (thermofisher.com). Primers and a probe that target the E gene to create a 113-base pair (bp) amplicon were used to detect SARS-CoV-2. Forward primer 5′-ACAGGTACGTTAATAGTTAATAGCGT-3′; probe 5′-FAM-ACACTAGCCATCCTTACTGCGCTTCG-BHQ1-3′, reverse primer 5′-ATATTGCAGCAGTACGCACACA-3′. Reaction concentrations of the SARS-CoV-2 primers and probe were 800 nM and 200 nM, respectively. An enterovirus exogenous PCR control (Asuragen, catalog number: 42050) was spiked in the RNA isolation step and the 61 bp amplicon was detected with the following primers and probe. Forward primer 5′-ATGCGGCTAATCCCAACCT-3′, probe 5′-VIC-CAGGTGGTCACAAAC-MGBNFQ-3′, and reverse primer 5′ CGTTACGACAGGCCAATCACT-3′ (VIC and MGBNFQ are proprietary dyes from Applied Biosystems. The reaction concentrations for the enterovirus primers and probe were 200 nM each. The AgPath-ID™ One-Step RT-PCR Reagents was used as per kit instructions (ThermoFisher, catalog number:4387391).): 5 μL of extracted RNA template was added to the master mix. Real-time PCR (RT-PCR) was performed on the Applied Biosystems 7500 Fast Real-Time PCR System thermocycler using the following amplification profile: one cycle of 50 °C, 30 min; one cycle of 95 °C, one min; 40 cycles of 95 °C, 15 s; and 60 °C, one min. Change in fluorescence was recorded at the elongation step of each cycle. Swabs with CT values ≥ 35.9 were considered negative. Samples above the cut-off point that did exhibit amplification were either retested at the Animal Health Center or sent to the National Center for Foreign Animal Disease in Winnipeg for confirmatory testing. CT values were not recorded at the AHC.

#### 2.5.2. SARS-CoV-2 PCR Testing at the BC CDC

All other swabs (live animal, environmental, and some mortalities) were refrigerated prior to shipment to the BC Centre for Disease Control (BC CDC) Public Health Laboratory, where they were tested for SARS-CoV-2 using qPCR. The BCCDC Public Health Laboratory (PHL) developed a real-time, multiplex reverse transcription PCR assay targeting the envelope (E) and the RdRP genes of SARS-CoV-2. The E gene primers and probes are as previously described above; the primer and probe details for RdRp are as follows: F-TGCCGATAAGTATGTCCGCA, R-CAGCATCGTCAGAGAGTATCATCATT, FAM-MGB-Probe-TTGACACAGACTTTGTGAATG. Briefly, total nucleic acids were extracted from 200 μL of specimen (swab/brush place in universal transport medium) using the Applied BioSystems MagMax™ Express 96 Nucleic Acid Extractor and the MagMax Viral/Pathogen Nucleic Acid Isolation Kit (Thermo Fisher Scientific, Waltham, MA, USA) according to the manufacturer’s recommendations. Real-time PCR was performed using the TaqMan Fast Virus Master Mix (Thermo Fisher Scientific) on the Applied Biosystems 7500 FAST real-time PCR system (Thermo Fisher Scientific). PCR set-up was performed using a volume of 5 µL of specimen extract, for a final reaction volume of 20 µL. Sample swabs with Ct values < 40 in both the E gene and RdRp were considered positive; those with Ct values < 40 in only one assay were considered indeterminant; and those with Ct values ≥ 40 were considered negative. For the purpose of this study, indeterminants were classified as negatives to improve specificity within our comparisons.

### 2.6. Statistical Analysis

Results were analysed with R, RStudio (2021.09.00, build 351), and Microsoft Excel (2020) software [18,19,20]. To compare the results of live animal vs. environmental samples, the number of samples and the percent positivity for each biweekly sampling period were displayed graphically. Error bars corresponding to 95% confidence intervals for the percent positivity were included on the graph. To compare the results of mortality vs. environmental samples, data were aggregated by month and compared monthly, as opposed to bi-weekly, given the low and variable mortality rate on farm. The number of samples, percent positivity, and error bars were produced and displayed similarly to the live animal vs. environmental sample graph. A kappa statistic was generated to compare percent agreement between the results of each live mink sampled and its corresponding cage. Mean CT values among positive samples for live animal and environmental samples were compared using a two-sample *t*-test. CT values were not compared between environmental samples and mortality samples because the majority of deceased mink were tested at the AHC where CT values were not recorded.

## 3. Results

### 3.1. Summary

In total, 1166 environmental samples (976 cage swabs; 190 manure trough swabs), 413 mortality swab samples, and 720 live swab samples were collected. There were 15 environmental sampling events, 12 live sampling events, and 42 mortality sampling events.

### 3.2. Environmental vs. Live Animal Sampling

At the level of the farm, when comparing PCR results (positive vs. negative) between live and environmental samples (including both cage and manure swabs) over time, trends in percent positivity of environmental samples generally mirrored that of live animal samples (Figure 1). However, the proportion of positive samples was consistently greater for environmental samples compared with live animal samples. Notably, there were two instances (2nd April 2021 and 28th January 2022) during the outbreak in which live animal samples were negative while environmental samples were positive. Environmental samples remained positive at an average of 1.69% positivity for the 6 weeks between depopulation and decommissioning of the farm (Figure 1). At the cage level, there was moderate agreement between PCR results for each live mink and its cage (Kappa = 0.45). Specifically, in 539/586 (92%) instances, mink and cage samples were in agreement (i.e., both positive for 23/586 (3.9%) and both negative for 516/586 (88.1%)). There were 28 instances in which the mink was negative, but its corresponding cage was positive; there were 19 instances in which the mink was positive, buts its corresponding cage was negative. Average CT values (Figure 2) for environmental samples (36.2, IQR = 2.1) were significantly higher than live animal samples (30.8, IQR = 7.6, t-stat= 2.685, *p*-value = 0.031).

### 3.3. Environmental vs. Mortality Samples

At the level of the farm, when comparing PCR results (positive vs. negative) between mortality and environmental samples (including both cage and manure trough swabs) over time, trends in percent positivity of environmental samples closely aligned with that of mortality samples (Figure 3). The number of mortalities submitted declined over the course of the study, and when the number of mortalities was relatively high (July and August) mortality samples had a higher proportion of positives than environmental samples, whereas the opposite was the case when mortalities were relatively low (September to March). CT values were not compared because the majority of mortalities were tested using serology at the AHC as opposed to the BC CDC.

### 3.4. Cage Swabs vs. Manure Trough Swabs

At the level of the farm, when comparing PCR results (positive vs. negative) between cage and manure trough swabs over time, trends in percent positivity of cage swab samples generally mirrored that of manure swabs (Figure 4). However, the proportion of positive samples was generally greater for manure trough swabs than cage swabs. The average CT value for cage swabs (36.99, IQR = 0.90) was not significantly different from that of manure trough swabs (36.29, IQR = 1.65, t Stat = 1.5, *p*-value = 0.153).

## 4. Discussion

The results of this study suggest that environmental samples could be used for the surveillance of SARS-CoV-2 outbreaks on American mink farms. At the level of the farm, trends in PCR results for environmental samples (positive vs. negative) mirrored those for both live animal and mortality samples over time. Furthermore, the percent positivity for environmental samples was generally greater than that of live animal samples, and there were two instances (2 September and 28 January) during the outbreak where live animal samples were negative while environmental samples remained positive. This may suggest that environmental samples are more ‘sensitive’ for the detection of SARS-CoV-2 infection circulating at a low prevalence in mink. Specifically, it is unlikely that the farm was truly negative on 2 September and 28 January, given that the subsequent sampling events found percent positivity values of 3.08% and 1.54%, respectively. The live animal sampling was performed in accordance with determining disease presence at a ≥5% prevalence, which does speak to the utility of using environmental sampling in parallel with live animal sampling. If live animal sampling was used alone, the farm may have been falsely declared negative in these two instances. Conversely, because viral cultures were not performed, it is possible that RNA detected were not viable virions, but instead persistent RNA contamination. However, environmental samples can detect disease at low prevalence. Wastewater surveillance had a detection probability of 87% when infection prevalence in the community was 0.01% for SARS-CoV-2 in humans [21].

Alternatively, in one Danish study where 100% of animals sampled on farm were positive, SARS-CoV-2 was only detected at low levels through air and water sampling, with no positives detected in the feed [16]. Similar results were seen in a Greek study, which found majority of the positive air and dust samples having low viral loads [7]. Another study, however, found high levels of SARS-CoV-2 in airborne dust during the testing of an infected farm, suggesting some variability with air sampling in the surveillance of farmed mink and the ability of environmental samples to detect disease at low or high prevalence [17]. In our study, the appearance of higher sensitivity of environmental samples may be due to more viral RNA accumulated on environmental surfaces over the sampling period relative to the amount present in the mink nasopharynx at a given point in time; however, further research would be needed to investigate this.

The high CT values of cage swab samples suggests that inactive viral RNA, rather than viable virions, was being detected. Previous research into the stability of SARS-CoV-2 in the environment found that infectious virus could remain on non-porous surfaces such as stainless steel for 7 days, whereas non-infectious viral RNA could be recovered for longer periods (up to 57 days) [22,23,24]. The persistence of inactivated viral RNA in the environment may also provide an explanation for those instances in which a live mink was negative while its cage was positive. Studies of ferrets, a close relative to American mink, have shown that peak viral shedding occurs between 2 and 4 days post infection and can decline below the limit of quantification by day 13; however, increased shedding was noted on days 16–18 [25]. This suggests some variability in the amount of virus shed by an individual, which could account for a positive cage swab when the individual mink within the cage tests negative. For those instances where the mink was positive and its cage was negative, it could be that the sampled cage surface was not contaminated with virus, or the viral RNA was present at a sufficiently low concentration to evade detection.

One of the most notable findings was the persistence of positive cage samples for at least 6 weeks after the mink had been depopulated. Less porous surfaces, such as steel, are known to absorb droplets less easily, preserving active virus for a period of time, which could explain this persistence [26]. The persistent detection could be inactive viral RNA, especially given the relatively high CT values. A similar finding was found in the study by Chaintoutis et al. where viral RNA was detected in dust two months after the resolution of the outbreak [7]. They posited that low temperatures during the winter months when the outbreak occurred may have created favourable conditions for viral preservation [7]. This is unlikely the case in our study because the persistent signals on our farm were detected in April.

Notably, cage swabs remained positive when the manure trough swabs became negative immediately after depopulation. This was unexpected given that wastewater studies have indicated that infectious particles have a relatively long persistence within sewage [27]. It may be the case that the manure collection system for the farm under study resulted in the rapid transit of manure through the trough apparatus; thus, trough samples provide a more contemporaneous picture of the herd infection status compared with cage swabs. This was demonstrated in wastewater surveillance for SARS-CoV-2, where sensitivity was affected by the time spent within the sewer network [21]. Another explanation could be the accumulation of organic debris on the cage surfaces (fur, feed, saliva, etc.) preserving the virus; organic matter has been known to stabilise other viruses in the environment [28]. This contrasts with the stainless-steel manure troughs that would likely be subjected to less static accumulation of organic matter.

Mink housing units have previously been used for sampling SARS-CoV-2, including testing food residues from the tops of cages, bedding material, water bowls and faecal material [17]. SARS-CoV-2 was detected in all of the housing unit sampling types, with the highest percentage of positive samples seen in bedding material and faecal material [17]. Faecal samples have also been evaluated with qt-PCR in tandem with serum ELISA and qt-PCR throat and nasal swab [10]. Among live mink sampled, serum ELISA was positive for 97% (29/30) of mink, with faecal samples only detecting viral RNA in 17% (5/30) of mink [10]. Conversely, dead mink faecal samples detected virus in 75% (3/4) of sampled individuals, with 100% of these individuals testing positive via nasal swabs [10]. This raises questions about when certain environmental sampling methods should be implemented during a SARS-CoV-2 outbreak, given that there appears to be variable sensitivity with the detection of disease.

When comparing both environmental sampling modalities, manure trough swabs had a consistently higher percent positivity compared with cage swabs, although trends in percent positivity over time were similar. This is consistent with the findings of a Danish study where 54% (51/95) of faecal material samples were positive compared with 10% (9/90) of the food residues swabs (similar to our cage swabs) [17]. This could indicate the higher viral load in faeces when compared with saliva/secretions and could explain why the live samples (essentially sampling secretions) would show as negative when the manure trough samples remained positive. The higher percent positivity of manure trough swabs may also be the result of the number of animals ‘represented’ by each sample type. While cage samples represent only the mink that was in that cage, manure trough swabs contain faeces from all mink within a row of cages. However, the Danish study tested faeces in an individual mink’s cage rather than pooled faeces, and faecal material samples were still superior. Therefore, it is unlikely that the discordance between manure trough swabs and cage swabs is solely due to the pooled nature of the manure trough swabs used in this study.

Our study suggests that environmental samples are not the preferred sample type for sequencing SARS-CoV-2 genomes obtained from mink farms. Cage and manure trough swabs had mean CT values of 36.99 (IQR = 0.90) and 36.29 (IQR = 1.65), respectively, which was significantly less than the CT values for animal samples (mean CT = 30.8, IQR = 7.6). For the assay used in this study, samples with CT values < 33 produced reliable sequencing results, with decreasing success as CT values increased. High CT values can be a result of low concentrations of RNA, and alternative sequencing methods should be investigated if phylogenetic trees are used in the surveillance for SARS-CoV-2 on mink farms. This is similar to what has been observed for the environmental surveillance of avian influenza virus, where researchers have turned to more advanced genomic techniques, such as targeted resequencing, to obtain AIV sequences from environmental samples [29].

An important limitation of this study is the fact that only one outbreak was included, which may limit the generalizability of our findings. Furthermore, no viral cultures were performed; thus, we could not determine whether positive PCR results reflected the presence of live virus vs. inactivated virus. Additionally, the AHC and BC CDC had different cut-off points for CT values to determine positive vs. negative. The lower CT value cut-off point at the AHC may have caused underestimates in mortalities that were positive compared with the live and environmental samples that were tested at BC CDC. However, this is unlikely to have altered the relative trends in positives overall because many of the mortalities were retested at the National Center for Foreign Animal Disease if amplification was detected above the cut-off point. Finally, because individual mink and cages were not re-sampled at subsequent time points, we could not determine how viral shedding or cage contamination in individual mink varied over time.

## 5. Conclusions

These data suggest that environmental samples could be reliable and sensitive for detecting and monitoring SARS-CoV-2 outbreaks in mink at the level of the farm, although their utility for the detection of infection in individual animals appears less consistent. Environmental samples may be best suited as a convenient method to survey seemingly negative farms and act as an early detection tool for SARS-CoV-2 outbreaks. Once an outbreak is determined, it would be advisable to continue environmental sampling in tandem with more conventional modalities to aid in the detection of low-prevalence infection within the farm. Environmental samples should not be used as the sole modality in determining disease freedom, given the questionable persistence post depopulation seen in our study. Additionally, it may be challenging to obtain whole genome sequences from environmental samples; however, this may be possible with different sequencing methods. Further research is needed to better understand SARS-CoV-2 persistence on different surfaces and substrates within the farm environment and to develop methods for obtaining sequences from environmental samples.

## Figures and Tables

**Figure 1 ijerph-20-01248-f001:**
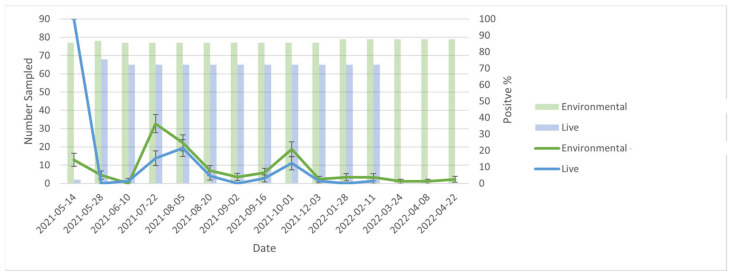
Comparison of environmental and live samples from 2021-05-14 to 2022-04-08. No live samples occurred after 2022-02-11 because the farm was depopulated.

**Figure 2 ijerph-20-01248-f002:**
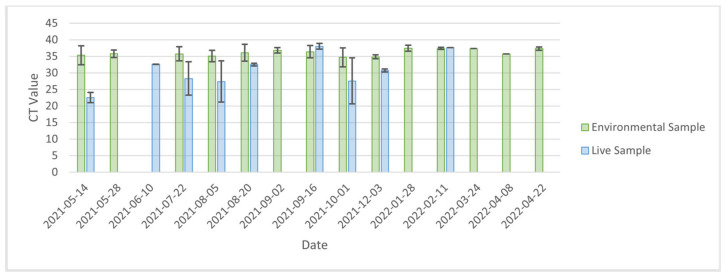
Comparison of average CT values for environmental and live animal samples. No live samples occurred after 2022-02-11 because the farm was depopulated.

**Figure 3 ijerph-20-01248-f003:**
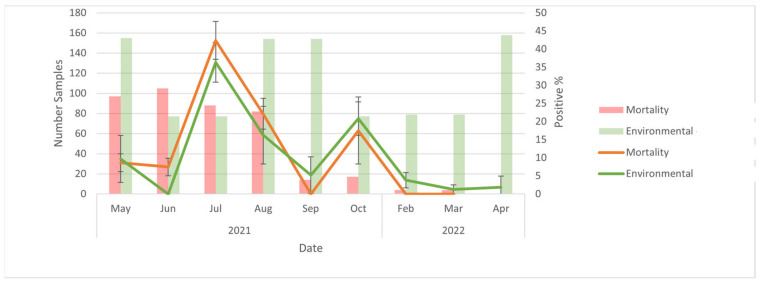
Comparison of environmental and mortality samples from May 2021 to April 2022. No mortalities were sampled after 2022-02-11 because the farm was depopulated.

**Figure 4 ijerph-20-01248-f004:**
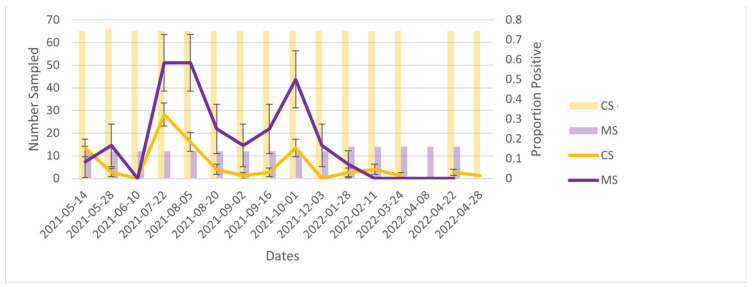
Comparing cage swabs (CS) and manure trough swabs (MS) throughout the outbreak. The farm was depopulated in April 2022.

## Data Availability

The data presented in this study are available on request from the corresponding author. The data are not publicly available due to privacy for the producers.

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
