# Peer review of "Hands off the Mink! Using Environmental Sampling for SARS-CoV-2 Surveillance in American Mink"

_ijerph, 2023, doi:10.3390/ijerph20021248_

Round 1

Reviewer 1 Report

I commend the authors on a very well-written paper and interesting, important study regarding the utility of environmental sampling for SCV2 surveillance on mink farms. I have several edits, suggestions and comments for the authors to consider. 

Abstract

Lines 16 and 18 - change mortality to deceased

Line 24 should read "environmental samples are viable sources"....

Keywords - authors can probably add another one unless there is a limit of 3

Introduction

Lines 29-42 - reference 1 is the only one used - is there possibly another reference to add as well to the first paragraph, maybe on wastewater surveillance efforts

Line 57 - "...within mink farms has historically been conducted...

Lines 70-85 - this section has such great information, but should be included in the discussion versus introduction. I suggest being brief here and stating something to the effect of .... "Previous studies have evaluated environmental sample types such as air, dust, bedding. Detection of SARS-CoV-2 in these samples was variable." Then go into the purpose of our study is to continue .... as you have in lines 88-90.

Lines 95-96 - I would rephrase question. You are not just looking at success of sequencing, but variant characterization. High Ct samples will not result in any useful genomic data to analyze. 

Methods

Line 98 - Suggest the subheading be "Study Location" or something similar since you are describing where the study took place and not how samples were being collected. Lines 118-119 could be moved here as well. 

Line 104 - remove "in" before infection on the first farm....

Section 2.2. - there is no mention of blood being collected here, but it is mentioned in the 2.5.2 and in the results 

Line 131 - live animal samples versus life

Line 145 - environmental sampling

Section 2.5.2 - be clear that blood was only collected in mortality instances. If collected from live mink, please indicate.

Line 181 - needs comma at the end

Line 187 - percent positivity - do not need 's

Line 191-192 - clean up sentence a bit to clarify. Perhaps state "Ct values were not compared between environmental samples and mortality cases in the mink, as majority of the deceased mink were tested using whole blood serology as opposed to PCR."

Lines 195-198 - be clear which sample types were used for which group (live versus deceased mink). For example, add swabs to line 197 (720 live swab samples).

Line 215 - higher versus more

Line 226 - delete "and thus did not have Ct values" - please scan for duplication on phrases. 

Section 3.5 - please ignore this comment if the journal requires figures to be formatting as such, otherwise, no need for subheading. 

Figures - please make them much bigger in the manuscript as whole - hard to read and the colors could be darker

Figure 1 - consider a log scale for y axis. 

All figures - I don't think you need to repeat language in the legend. For example, have environmental and live next to the colored lines in the legend will suffice - do not need - environmental - number sampled, etc. Or keep in first graph and remove later on. 

Figure 5 - written text will suffice - I do not think you need a figure as well. 

Discussion

Lines 270-274 - You include so much previous data in the introduction on previous environmental sampling studies for SCV2. I highly recommend using those references here in the discussion and removing mention of M. avium. This is a SCV2 focused paper - there is no need to introduce another pathogen when you have so much other data. 

Overall - congratulations to the authors on an exceptional paper. I do not see much mention of genomic sequencing on the positive PCR samples from the live mink. Please discuss this. 

Beyond what it outlined above, I have no further comments for the authors. This paper is very timely as the science community continues to pursue surveillance efforts for SCV2 across the human-animal-environmental interface. 

Reviewer 2 Report

Manuscript Number: ijerph-2104890

Title: Hands off the Mink! Using Environmental Sampling for SARS-CoV-2 Surveillance in American Mink

Authors: Boyd et al.

Journal: IJERPH (ISSN 1660-4601)

Original Article Submission Recommendations: Revisions Requested

Comments to the Authors

1.       Overview and General Recommendation: Overall, this is an interesting and relevant discussion of the utility of environmental sampling in the early detection of SARS-CoV-2 outbreaks in farmed mink. The authors had a unique opportunity to study live animal, dead animal, and environmental samples through a regional outbreak in farmed mink. PCR was the primary method utilized. This study found that environmental sampling is very similar to that obtained from live and dead animals. While the precise reason for increased environmental sensitivity in PCR testing is unclear, environmental PCR testing shows promise as a method of early outbreak detection in farmed mink.

2.       Major Comments (Revisions Requested)

2.1.    Please edit for grammar, punctuation, verb tense and identify run-on sentences.

2.2.    While you discuss statistical methods, no statistical data is included on the tables or graphs and needs to be added.

2.3.    Adjust the figures to be incorporated into the text as they are discussed in the results section.

2.4.    The discussion implies that environmental sampling may be more sensitive to picking up on lower prevalence infection than live animal sampling, but you really need to address the longevity of the RNA in the environment with an outbreak. You haven’t ruled out that the RNA is simply being detected at high CT values in the environment after infection is cleared and that this detection of RNA (not viable virus, but just the RNA) persists. I know that you go on to address this is the next paragraph, but the initial paragraph (page 8, lines260-276) gives the impression that environmental PCR will detect low prevalence disease as opposed to detecting persistent contamination.

2.5.    It’s unclear when you are evaluating based on serology in your reported figures? Please address how you are using these data to confirm that you are not comparing serology positives to PCR positives.

2.6.    You are missing several key references that address similar topics: Chaintoutis et al (Plos Pathog 2021), Lu et al (Nature Communications 2021) are two examples; additionally, there are a few references that seem irrelevant (Microsoft Excel, for example)

3.       Minor Comments (Revisions Encouraged)

3.1.    Introduction: Please consider addressing what we know about environmental surveillance for SARs-CoV-2 in people as well.

3.2.    Materials and Methods 2.1. Sample Collection: This section does not actually address the types and methods for sample collection. It attempts to provide a timeline for the outbreak, but the wording is confusing as well. Line 104: “The course of in infection” is a typo; what does “largely subclinical within only relatively small numbers of mortalities” mean? Do you mean to say that the first outbreak was mostly subclinical with very few fatalities? What about the second farm?

3.3.    Materials and Methods 2.2. Animal Surveillance: is there a reference for this method oropharyngeal of sampling in mink? What do we know about sensitivity or specificity?

3.4.    Results 3.2. How do you define a positive and negative PCR result? Is there a high CT cut-off value that is considered negative? Or is any detection, regardless of CT value, considered positive?

3.5.    Results 3.5. Figures: Please insert figures and tables into the results section as discussed and not into their own section.

3.5.1. Figure 1: Adjust y-axis to incorporate the data points. I did see that you explain it in the legend, but it’s still a little misleading to not include it in the data.

3.5.2.Table 1: leave this out and just report predictive values, % positivity etc

3.5.3.Figure 3 line 246-247: mortalities, not moralities

3.6.    Discussion, line 270-272: there is no need to compare to M avium here. Very different pathogens.

3.7.    Please define “disease freedom” – I’m not familiar with this term, but is this a standard method for several diseases or just “free of SARS-CoV-2 infection”?
